# Childhood Acute Lymphoblastic Leukemia Showing Unilateral Motor Dysfunction Prior to Chemotherapy: A Diffusion Tensor Tractography Study

**DOI:** 10.3390/children10020224

**Published:** 2023-01-27

**Authors:** Jong Bum Kim, Jae Min Lee, Su Min Son

**Affiliations:** 1Department of Physical Medicine and Rehabilitation, College of Medicine, Yeungnam University, Daegu 42415, Republic of Korea; 2Department of Pediatric Medicine, College of Medicine, Yeungnam University, Daegu 42415, Republic of Korea

**Keywords:** leukemia, diffusion tensor, motor, chemotherapy

## Abstract

This study aimed to evaluate children with lymphoblastic leukemia and examine the potential correlation between corticospinal tract (CST) injury and motor dysfunction prior to chemotherapy using diffusion tensor tractography (DTT). Nineteen consecutive patients with childhood leukemia (mean age 7.483 ± 3.1 years, range 4–12 years) with unilateral motor dysfunction who underwent DTT prior to chemotherapy and twenty healthy individuals (mean age 7.478 ± 1.2 years; range 4–12 years) were enrolled. Motor functions were evaluated by two independent investigators. The cause of neurological dysfunction was identified based on the CST state using mean fractional anisotropy (FA), mean fiber volume (FV), and CST integrity using DTT. All patients showed disrupted integrity and significantly decreased FA and FV in the affected CST compared to the unaffected CST and the control group (*p* < 0.05). These DTT results also corresponded to patients’ unilateral motor dysfunction. Using DTT, we demonstrated that neurological dysfunction may occur in patients with childhood acute lymphoblastic leukemia even prior to chemotherapy, and that CST injuries correlate with motor dysfunction in these patients. DTT may be a useful modality for evaluating the neural tract state in pediatric leukemia patients with neurological dysfunction.

## 1. Introduction

Acute lymphoblastic leukemia (ALL) is the most common type of childhood leukemia that comprises up to 75–80% of all leukemia diagnoses [1]. Patients with ALL have more neurological deficits than those with other types of leukemia [2,3]. Khan et al. reported that neurological deficits were present in 83% of long-term childhood ALL survivors [3]. Various neurological complications, such as dizziness, headache, seizures, cognitive dysfunction, and motor dysfunction have been reported in patients with childhood ALL [3,4,5,6]. An exact and detailed evaluation of the patients is very important for the determination of a therapeutic plan. When diagnosis is delayed or missed, immediate and proper treatment will be delayed, which may result in poor prognosis [7,8]. However, it is often difficult to detect neurological deficits in patients with leukemia because of their poor physical condition, and the tendency to bed-rest rather than engage in active outdoor activities. Furthermore, even when neurological problems are recognized, detailed clinical evaluation is limited because of poor patient compliance, low functional level, and poor medical condition [7,9].

All children with ALL get CNS-directed treatment because patients who experience CNS recurrence have a very poor prognosis, and since there is currently no reliable way to identify individuals who have CNS involvement or patients who are at high risk of experiencing CNS relapse. In addition to patients with CNS involvement at diagnosis, traumatic lumbar puncture at diagnosis also adversely affects outcome in childhood acute lymphoblastic leukemia [10]. The evaluation of cytomorphology on CSF cytospin slides is the current gold standard for identifying CNS involvement. Due to its poor reproducibility and sensitivity, this method is ineffective. Flow cytometric measurement (FCM) of CSF has recently attracted attention as a unique, highly specific, and sensitive technique for detecting leukemic cells in the CNS [11]. 

Another issue regarding the recognition of neurological deficits in patients with childhood ALL is whether their neurological dysfunctions occur only with chemotherapy. It is currently known that chemotherapy is related to neurological deficits in patients with leukemia [12]. In previous studies on childhood leukemia, it has been suggested that leukemia involves the central nervous system (CNS) and causes neurological deficits [12]. A few reports on CNS infiltration in patients with ALL prior to chemotherapy have been published. In 2013, Laperche et al. examined the cerebrospinal fluid (CSF) in 108 patients with childhood ALL prior to chemotherapy and reported abnormal findings in the CSF, and CNS infiltration in 14% of the patient samples [13]. Another report demonstrated the presence of leukemic blasts in the CSF in one-third of the patients with ALL prior to chemotherapy [14].

These previous studies support the hypothesis that neurological deficits may occur through CSF infiltration in patients with ALL, regardless of chemotherapy. However, these studies only assessed CSF infiltration, and neural tract injury associated with neurological deficits was not assessed in children with ALL.

The recently developed diffusion tensor tractography (DTT), which derives from diffusion tensor imaging (DTI), enables evaluation of the integrity of white matter tracts in three dimensions by imaging water diffusion characteristics [15,16]. DTT has been used in several studies to evaluate the corticospinal tract (CST) for motor function in children [17,18,19]. We examined patients with childhood ALL, using DTT to determine whether neurological deficits occurred prior to chemotherapy, and whether there were CST injuries corresponding to the neurological deficit.

## 2. Materials and Methods

### 2.1. Participants

Nineteen children (mean age 7.483 ± 3.1 years, range 4–12 years) with ALL were enrolled according to the following criteria: (1) diagnosis of ALL, (2) age under 13 years, (3) DTT before induction of chemotherapy, (4) unilateral motor dysfunction, (5) no definite abnormal findings on conventional brain magnetic resonance imaging (MRI), (6) no specific perinatal history, including preterm birth, (7) no serious traumatic events or operation history of the musculoskeletal system and brain, and (8) no history of developmental delay. Patients born prematurely were excluded due to the possibility of microscopic brain lesions related to motor function. Patients with brain damage, musculoskeletal damage, or delayed development were also excluded because it was not possible to distinguish whether damage was caused due to the leukemia. In addition, patients with bilateral motor dysfunction were excluded from this study, because of the possibility that motor dysfunction was due to general weakness caused by leukemia, thereby hindering examination of a potential direct association between neurological deficits and clinical symptoms. Initially, 42 patients with ALL who underwent DTT were recruited in this study. Twelve patients who underwent DTT after chemotherapy were excluded from this study; seven patients with bilateral motor dysfunction were excluded from this study; two patients with preterm birth and one patient with delayed development were also excluded. Another patient was excluded because of a history of traumatic brain injury. Finally, 19 patients were enrolled in the study. Additionally, we classified the patients’ CSF state as follows: CNS 1, no detectable leukemic cells in the CSF; CNS 2, less than 5 white blood cells per microliter of blood and a positive “cytospin” for blasts; and CNS 3, more than 5 white blood cells per microliter of blood and a positive “cytospin”. A traumatic lumbar puncture was defined as more than 10 red blood cells per microliter of CSF [20]. 

Twenty age-matched, healthy control individuals (mean age 7.478 ± 1.2 years, range 4–12 years) without any specific history of delayed development or neurological problems were enrolled in this study. This study was conducted prospectively, and parents of all the participants provided informed consent. The study was approved by our institutional review board.

### 2.2. Functional Evaluation

All patients were clinically evaluated for motor function at the time of ALL diagnosis, and before chemotherapy or cranial irradiation. Manual muscle testing (MMT) of the upper and lower extremities was performed to evaluate motor weakness. Depending on the ability of patients to be examined, grip strength test for the assessment of hand strength, and Purdue pegboard test (PPT) for the assessment of fine motor coordination skills of the hands were performed. Patients, who were not able to perform these examinations because of poor condition, were interviewed in detail with their parents using relevant questionnaires to determine their motor function. The questionnaire aimed to understand whether the hand dominance had changed, whether the previously good motor function had become poor, or whether motor function showed abnormal patterns that were not present earlier. Clinical evaluation of the patients was performed independently by two pediatric neurologists who were blinded to each other’s evaluation and DTT results. We excluded patients who showed disparate results between the two evaluators. The results of the two evaluators for all nineteen patients were consistent; therefore, all patients were included in this study.

### 2.3. Diffusion Tensor Imaging

DTI was performed on the day of, or 1 day after, clinical evaluation. DTI data were acquired using a 1.5 T Gyroscan Intera system (Philips Healthcare, Best, The Netherlands) equipped with a sensitivity encoding head coil using a single-shot spin-echo-planar imaging sequence. We acquired 60 contiguous sections parallel to the anterior/posterior commissure line for each of the 32 noncollinear and noncoplanar diffusion-sensitizing gradients. The following imaging parameters were applied: matrix 128 × 128, field of view 221 × 221 mm^2^, time of echo 76 ms, time of repetition 10,726 ms, sensitivity encoding factor 2, EPI factor 67, b 1000 s/mm^2^, NEX 1, and section thickness 2.3 mm.

The Oxford Centre for Functional Magnetic Resonance Imaging of the Brain software (www.fmrib.ox.ac.uk/fsl, accessed on 14 November 2022) was used to preprocess the DTI datasets. Eddy current–induced image distortions and motion artifacts were removed using affine multiscale two-dimensional (2D) registration. DTI studio software (Center for Magnetic Resonance Microimaging; Johns Hopkins University, Baltimore, Maryland) was used for CST evaluations. Fiber tracking was performed using the fiber assignment continuous tracking algorithm, brute-force reconstruction, and multiple-region of interest (ROI). A seed ROI was drawn on the CST region in the anterior mid pons on several 2D fractional anisotropy (FA) color maps of each patient. The target ROI was drawn on the CST portion of the anterior lower pons. Fiber tracts passing through both ROIs were designated as final tracts of interest. Termination criteria used for fiber tracking included an FA of 0.2 and an angle change of 45°. We evaluated and compared diffusion parameters, such as mean FA, mean fiber volume (FV) of entire CSTs for the right and left hemispheres, and CST integrity that passed through the mid pons to the cortex in patients and control individuals. In healthy controls, no definite difference was observed between the right and left sides; therefore, the mean FA and FV data for both sides were used for comparison with those of the patient cohort.

DTI data from each participant were analyzed by two authors, and considered reliable when the results of CST integrity were consistent. There was no discrepancy between results; therefore, all DTT data from the 19 participants were included in this study.

### 2.4. Statistical Analysis

Statistical analysis was performed using the following steps. First, the Kolmogorov–Smirnov test was used to test the normality of each group, and it was confirmed that all significance probabilities were 0.05 or more with a normal distribution. Second, Fisher’s exact test was used to compare the demographic data between groups. Third, we used an independent *t*-test to assess the significance of differences in CST parameters, such as FA and FV, between the affected and unaffected sides of the patients. Finally, the differences in CST parameters between the patient and control groups were assessed using an independent t-test. In the control group, the mean FA and FV values including both the right and left CSTs were used. Statistical significance was set at *p* < 0.05, and analysis was performed using the statistical package for the social sciences, version 18.0 (SPSS, Chicago, IL, USA).

## 3. Results

### 3.1. Demographic and Functional Data

The demographic data of the patient and control groups are shown in Table 1. No significant difference was observed in the demographic data between the two groups, except for in the history of intrathecal chemotherapy and cranial irradiation. All evaluations including clinical function and DTT were performed before chemotherapy or cranial irradiation. The CSF state was assessed after initiation of chemotherapy (mean 4.36 ± 3.2 days, range 1–14 days). Three out of nineteen patients (15.8%, two patients with CNS 2 and one patient with CNS 3) had detectable leukemic blasts in the CSF without bleeding. Bleeding into the CSF was noted in two patients (10.5%) because of traumatic lumbar puncture.

Out of 19 patients, 6 complained of gait disturbance, and 3 had poor hand function. Patients who complained of gait disturbance reported that they could not walk properly, because of limping, and that they easily fell during walking. In patients with gait disturbance, the one-leg standing time was different between the two sides, and unilateral weakness was diagnosed by MMT on the right and left sides. Three patients who complained of poor hand function showed unilateral motor weakness on MMT of both extremities. In the questionnaire on motor function, four patients reported a change in hand dominance. Eight patients showed meaningful differences in PPT between the right and left sides, even when considering the difference between dominant and nondominant hands. Among these eight patients, five showed better results in the nondominant hand than in the dominant hand. In the control group, MMT and PPT were not significantly different between the right and left sides.

### 3.2. Diffusion Tensor Tractography

All CSTs in control individuals showed preserved integrity in both hemispheres. In patients, unaffected CSTs showed preserved integrity from the midbrain to the cortex level, whereas affected CSTs showed disrupted integrity (Figure 1). Diffusion parameters for the CST analysis are listed in Table 2. In the patient group, FA and FV were significantly lower in the affected side than in the unaffected side (*p* < 0.05). Comparative analysis between patient and control groups showed significant differences in the FA and FV values. When comparing diffusion parameters between the affected CST in the patient and control groups, the patient group showed significantly decreased FA and FV compared to those of the control group (*p* < 0.05). In the comparison between the unaffected side between patients and control individuals, patients also showed significantly decreased FA and FV (*p* < 0.05). However, the difference was prominent in the comparison between the affected side and the control group compared to the unaffected side and controls.

## 4. Discussion

In the current study, through detailed clinical and DTT evaluations, we found that all patients with ALL with motor dysfunction showed corresponding DTT findings, including disrupted integrity and decreased DTT parameters of CS even prior to chemotherapy. Patients with ALL may have neurological deficits in the CNS regardless of chemotherapy, owing to the following reasons. First, clinical motor dysfunction was observed in all patients prior to chemotherapy. Second, DTT parameters, such as FA and FV, of the CST of the hemisphere contralateral to the motor dysfunction in patients were significantly decreased compared to those in the opposite side or in the control group. FA indicates white matter integrity (e.g., myelination, axon diameter, fiber density, and fiber organization) and FV indicates the number of voxels within a neural tract [21,22]. It is well known that affected tracts show decreased FA or FV values [23,24]. Therefore, these results suggest that the CST of the hemisphere contralateral to the motor dysfunction was affected in our patients. Additionally, our DTT results showed that the integrity of the affected CST was disrupted in all patients with corresponding unilateral motor dysfunction. However, the unaffected CSTs in patients and the CSTs in control individuals showed preserved integrity and descended along a known CST pathway. Finally, the comparison of diffusion parameters between patients and the control group showed that the affected CST in patients showed more prominent differences than the unaffected CST. These results imply that the disrupted CST states in our patients were pathological injuries related to motor dysfunction, rather than brain immaturity. In addition, they imply that unilateral motor dysfunction in our patients may be mainly attributed to injury in the corresponding CST.

The cytopathological analysis of a CSF cytospin smear is the current method for determining whether ALL is present in the CNS. This method’s sensitivity is lower than that of PCR-based or flow cytometry-based approaches [25,26]. The National Comprehensive Cancer Network advises utilizing FCM in the diagnosis of CNS lymphomas because it is more sensitive than traditional cytology at identifying tumor cells [27,28]. A positive result in the detection of minimal residual disease during the treatment is strongly associated with a worse prognosis, and it may be an indication for a more intensive therapy [29,30]. Both FCM and PCR techniques have been thoroughly applied for detecting the levels of MRD in bone marrow aspirates, becoming crucial elements in the management of leukemias. When using FCM or PCR techniques, many children with ALL had subclinical disease in the CNS at the time of diagnosis [25,31,32]. Occult/subclinical CNS leukemia’s clinical significance is still unknown.

A number of high-risk factors, such as infant age, T-cell lineage, hyperleukocytosis, and specific chromosomal aberrations in B-ALL, like KMT2A gene rearrangements, t(9;22)[BCR-ABL1] and t(1;19)[TCF3-PBX1] are linked to CNS involvement [33]. Early in the course of the disease, leukemic cells spread to the CNS, and with current diagnostic tests, CNS involvement is seen in around 10% of patients at diagnosis [34,35,36,37].

However, the actual frequency of CNS involvement is probably higher due to the shortcomings of existing diagnostic methods. This is corroborated by the observation that post-mortem brain biopsies frequently reveal considerable CNS infiltration in patients who do not have cells in their CSF [10]. At diagnosis, CNS involvement is linked to worse mortality and a higher risk of CNS relapse [36]. Although the majority of CNS relapses are observed in individuals who were CNS-negative at diagnosis, this highlights the limitations of the existing CNS classification [36,38].

Additionally, not all individuals with CNS positivity upon diagnosis go on to experience a CNS relapse. All patients with ALL get CNS-directed therapy, which includes intrathecal chemotherapy, systemic high-dose chemotherapy and, in certain cases, cranial irradiation, due to the low prognostic value of existing CNS classifications [33]. These treatments increase the risk of long-term cognitive and neuroendocrine deficits by causing acute and chronic neurotoxicity. Leukocyte counts, CSF cytospin results, and occasionally, clinical observation of the disease’s symptoms are used to make the diagnosis of CNS leukemia. If at least one blast cell is visible on cytospin, CNS leukemia is identified.

CSF flow positive at diagnosis varied from 16.3% to 40.5% in studies with ALL cohorts, compared to 2.8% to 18.1% for cytospin [11,13,39,40,41,42]. Blasts can be detected in FCM even in patients with negative findings in the CSF cytospin test, which means that a small amount of blast cannot be detected in the cytospin test.

Symptoms of CNS involvement of leukemia included headache, nausea/vomiting, seizure, loss of consciousness, confusion, anxiety, gait abnormalities, visual changes. Several studies [34,43,44] have linked CNS3 disease to an elevated risk of CNS relapse and poor outcomes; however, the link between CNS2 and later CNS relapse is less consistent [34,36,43]. Since there have been significant variabilities in CNS2 rates, variable cytospin methods across studies are more likely to account for these disparities than clinical variations. The majority of CNS relapses occur in CNS1 patients, despite the fact that CNS3 and CNS2 have been described as predictive indicators of CNS recurrence in several studies, which emphasizes the need for new biomarkers for CNS leukemia [36,38].

A traumatic lumbar puncture (TLP) is when the CSF is artificially contaminated with blood (≥10 erythrocytes per μL CSF) while being punctured. Childhood ALL TLP has previously been linked to CSF blasts and a higher risk of CNS relapse [10,36,43]. The biological connection between iatrogenic leukocyte introduction and the risk of relapse is still unclear. The presence of leukemic blasts in the CSF was validated by flow cytometry in a recent study, which revealed that TLP upon diagnosis was only linked to a higher risk of relapse in patients [11].

Previous studies have been conducted in patients with ALL with motor dysfunction. In 2005, a report described balance problems and motor weakness in children with leukemia during chemotherapy [45]. Another study by Hartman reported impaired motor performance in patients with ALL after completion of chemotherapy [46]. In 2013, Green et al. reviewed 28 studies and reported that patients with childhood ALL have poorer gross and fine motor performance during treatment than healthy individuals [47]. However, there was no other evaluation for specific neurological tract injuries associated with clinical symptoms.

Some previous DTI studies on white matter injury in patients with childhood leukemia have been conducted [48,49,50,51,52]. Porto et al. performed DTI scanning in adult long-term survivors of acute childhood leukemia 13.9 years after diagnosis, and compared FA values for the entire white matter region between patients and controls. Their results showed white matter differences with decreased FA in patients, even without any changes in conventional T2-weighted imaging [48]. However, they did not evaluate any clinical functions and did not determine any specific neural tract injuries. Another DTI study on childhood ALL was conducted by Edelmann et al. They performed a comparative DTI study between ALL survivors (mean age 24.9 years) 15.0 years after ALL diagnosis and healthy controls. They assessed neurocognitive functions, including motor speed. Their results showed that patients had poor performance compared to controls, and that clinical performance was significantly associated with altered neuroanatomical integrity in patients [49]. In 2016, Cheung et al. reported a DTT study on survivors of childhood ALL with leukoencephalopathy during treatment. They performed an initial brain MRI at a mean time of 7.7 years after diagnosis and performed follow-up brain MRI, including DTI, in patients diagnosed with acute leukoencephalopathy at the initial MRI scanning. They showed decreased DTI parameters and associated clinical dysfunction of executive function [52]. All those studies performed DTI several years after ALL treatment and showed decreased DTI parameters and associated motor function problems. With aging, the white matter of the brain undergoes significant changes, including maturation or degeneration. Therefore, evaluation of the white matter that was performed several years after ALL treatment may hinder the diagnosis of changes caused by treatment of the leukemia, and changes in white matter caused by the leukemia itself are even more challenging to be determined. 

We reported a case series for the disruption of the CST in patients with acute ALL [53]. However, it was not clear whether the white matter injury was caused by ALL itself or by neurotoxicity-related chemotherapy. Here, to reduce bias and confirm a clear association between clinical motor dysfunction and neurological deficit, we prospectively recruited patients according to strict criteria. We excluded patients with brain lesions on conventional brain MRI to ensure a lack of existing neurological deficits unrelated to leukemia. Additionally, we only recruited patients with unilateral motor dysfunction to exclude patients with motor dysfunction resulting from general weakness. Finally, to eliminate bias, two evaluators were blinded, and participants were enrolled only when the analysis was consistent between the two evaluators. We tried to perform an objective evaluation as long as the patients’ condition allowed it, in addition to interviews with questionnaires. In the current study, we demonstrated that leukemia itself can affect the CNS and lead to clinical neurological dysfunction even prior to chemotherapy. To the best of our knowledge, this is the first study to identify CST injury corresponding to clinical motor dysfunction in patients with childhood ALL prior to chemotherapy. 

However, this study had some limitations. First, the sample size was small. Second, the broad age range, variability in clinical symptoms, and compliance issues with different medical conditions hindered the performance of identical clinical measurements, and the assessment of statistical significance for clinical motor dysfunction. Third, it was not possible to examine the CSF through cytopsin or the FCM technique to confirm the presence of leukemic blasts to minimize the risk of CNS contamination due to traumatic tapping. At the time of leukemia diagnosis, there are many blasts in the blood. To minimize the risk of CNS contamination due to traumatic tapping, we performed intrathecal chemotherapy and CSF examination when no blast was observed in peripheral blood. Finally, DTT may underestimate or overestimate neural tract injuries because of regions of tract complexity. Therefore, further studies with a larger number of participants and identical evaluation tools are required. 

In conclusion, we demonstrated that patients with childhood ALL may have neurological deficits, and that patients have CST injury corresponding to their motor dysfunction even prior to chemotherapy. We believe that the application of DTT could be helpful toevaluate patients with childhood leukemia characterized by neurological deficits.

## Figures and Tables

**Figure 1 children-10-00224-f001:**
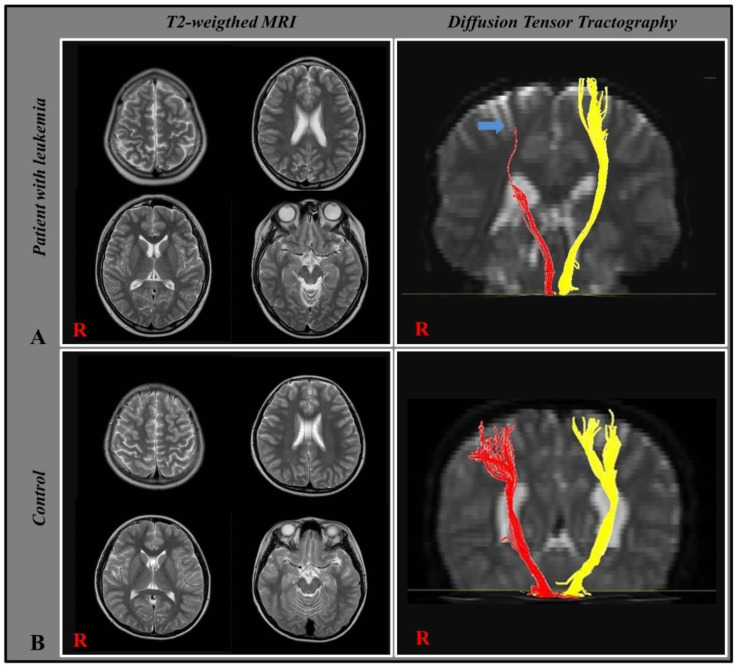
Conventional brain magnetic resonance imaging (MRI) and diffusion tensor tractography (DTT) results. The patient had no abnormal findings on conventional brain MRI despite definite unilateral motor weakness in the left hand. However, DTT shows disrupted integrity of the right corticospinal tract at the corona radiata level (blue arrow) corresponding to patient’s left motor weakness. Healthy individuals showed no abnormal findings on conventional brain MRI or DTT of the corticospinal tract. R means right, this is common notation in brain images, self explanatory. Red and yellow color were used to distinguish right from left side.

**Table 1 children-10-00224-t001:** Demographic data of participants.

		Patients Group (n = 19)	Control Group (n = 20)	*p*-Value
Age (years)		7.483 ± 3.1	7·534 ± 4.12	0.845
Sex (male: female)		16:3	17:3	0.607
Gestational age at birth (weeks)		39.6 ± 1.2	38.7 ± 2.1	0.135
Birth weight (kg)		3.6 ± 0.7	3.8 ± 0.5	0.264
Hand dominance (n, Right/left)		19 (14:5)	20 (16:4)	0.531
Acute leukemia type (B-cell/T-cell)		19 (17:2)		
Intrathecal chemotherapy		19		0
Cranial irradiation		2		0
Duration of chemotherapy and CSF study (days)		4.36 ± 3.2		
CNS state	CNS 1	14 (73.6%)		
	CNS 2	2 (10.5%)		
	CNS 3	1 (5.3%)		
	Traumatic lumbar puncture	2 (10.5%)		

Values are expressed as the mean ± standard deviation. Significance was set at *p* < 0.05.

**Table 2 children-10-00224-t002:** Diffusion parameters of the corticospinal tract.

CST	Patients Group (n = 19)	Control Group (n = 20)	*p*-Value
FA			
Affected side	0.309 ± 0.052	0.579 ± 0.05	0.001 *
Unaffected side	0.528 ± 0.053	0.579 ± 0.05	0.005 *
*p*-value	0.003 *	-	-
FV			
Affected side	289.8 ± 100	1081.8 ± 171	0.001 *
Unaffected side	869.6 ± 131	1081.8 ± 171	0.002 *
*p*-value	0.005 *	-	-

Values are expressed as the mean ± standard deviation. Significance was set at *p* < 0.05 (*). FA = fraction anisotropy; FV = fiber volume; CST = corticospinal tract.

## Data Availability

All data are provided in the paper.

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
