# Peer review of "Childhood Acute Lymphoblastic Leukemia Showing Unilateral Motor Dysfunction Prior to Chemotherapy: A Diffusion Tensor Tractography Study"

_children, 2023, doi:10.3390/children10020224_

Round 1
Reviewer 1 Report
line 44-48: reference 11 correspond to Evans EA (not Laperche et al)
line 67-68: please try to explain why perinatal history represents an excluding criteria
line 78-79; SUGGESTION: Additionally, we performed CSF study to classify patients' CSF state as follows:
line 105: "DTI was performed on the day of or 1 day after the cjlinical evaluation". Here you need to state that clinical evaluation was performed before intrathecal chemotherapy or radiotherapy.
line 148: The link provided here sent me to a table that includes 45 cubjects distributed in 3 groups; please advice and correct
line 149-150: One essential criteria was DTT befores induction of chemotherapy. You must clearly state here that intrathecal chemotherapy and/or irradiation were administered after functional evaluation and DTT. Alternatively, if this is the case, I would suggest to omit data regarding intrathecal chemotherapy and cranial irradiation.
line 170: I could not find figure 1 within the original manuscript nor the supplementary files
line 255-256: In the section 3.1 you do describe CSF findings. Please clearly state here that you could not examine the CSF BEFORE functional evaluation and DTT
Author Response
- line 44-48: reference 11 correspond to Evans EA (not Laperche et al)
Answer: Thank you very much for your comment. I revised the sentence as follows.
There were other reports on CNS infiltration in patients with ALL prior to chemotherapy. In 2013, Evans et al. examined the cerebrospinal fluid (CSF) of 108 patients with childhood ALL prior to chemotherapy
- line 67-68: please try to explain why perinatal history represents an excluding criteria.
Answer: I appreciate your comment. I revised the text as follows.
…7) no serious traumatic events or operation history of the musculoskeletal system and brain, and 8) no history of developmental delay. Patients born prematurely were excluded due to the possibility of microscopic brain lesions related to the motor function. Patients with brain damage, musculoskeletal damage, or delayed development were also excluded because it was difficult to distinguish them from damage caused by leukemia itself.
- line 78-79; SUGGESTION: Additionally, we performed CSF study to classify patients' CSF state as follows:
Answer: Sorry that I didn't understand the meaning of this. Please add more explanation, then I will revise the text following your opinion.
- line 105: "DTI was performed on the day of or 1 day after the cjlinical evaluation". Here you need to state that clinical evaluation was performed before intrathecal chemotherapy or radiotherapy.
Answer: I agree with your opinion. I revised the text as follows.
All patients were clinically evaluated for motor function at the time of ALL diagnosis and before chemotherapy or cranial irradiation.
- line 148: The link provided here sent me to a table that includes 45 cubjects distributed in 3 groups; please advice and correct.
Answer: Sorry for that. Maybe it was a systemic error. I included the table as follows.
Table 1. Demographic data of subjects
|
Patients group |
Control group |
p-value |
||
|
Age(years) |
7·483±3·1 |
7·534±4·12 |
0·845 |
|
|
Sex(male:female) |
16:03 |
17:03 |
0·607 |
|
|
Gestational age at birth(weeks) |
39·6±1·2 |
38·7±2·1 |
0·135 |
|
|
Birth weight(kg) |
3·6±0·7 |
3·8±0·5 |
0·264 |
|
|
hand dominance (n, Right/left) |
19(14:5) |
20(16:4) |
0·531 |
|
|
acute leukemia type |
19(17:2) |
|||
|
Intrathecal chemotherapy |
19 |
0 |
||
|
Cranial irradiation |
2 |
0 |
||
|
Duration of chemotherapy and CSF study(days) |
4·36±3·2 |
|||
|
CNS state |
CNS 1 |
14(73·6%) |
||
|
CNS 2 |
2(10·5%) |
|||
|
CNS 3 |
1(5·3%) |
|||
|
Traumatic lumbar puncture |
2(10·5%) |
|||
Values are expressed as the mean ± standard deviation. Significance was set at P-value < 0.05.
- line 149-150: One essential criteria was DTT before induction of chemotherapy. You must clearly state here that intrathecal chemotherapy and/or irradiation were administered after functional evaluation and DTT. Alternatively, if this is the case, I would suggest to omit data regarding intrathecal chemotherapy and cranial irradiation.
Answer: Thank you for your comment. The timing of evaluation such as clinical performance and DTT is very important for this study. I added the sentences in the results section as follows.
All evaluations including clinical function and DTT were performed before chemotherapy or cranial irradiation. For the CSF study, it was performed after the initiation of chemotherapy (mean 4.36±3.2 days, range 1–14 days).
- line 170: I could not find figure 1 within the original manuscript nor the supplementary files.
Answer: Sorry about this. It seems that there was a systemic error. I included figure in the main text as below.
Figure 1. Conventional brain magnetic resonance imaging (MRI) and diffusion tensor tractography (DTT) results. The patient had no abnormal findings on conventional brain MRI despite definite unilateral motor weakness in the left hand. However, the result of DTT shows disrupted integrity of the right corticospinal tract at the corona radiata level (blue arrow) corresponding to his left motor weakness. The control subjects showed no abnormal findings on conventional brain MRI or DTT of the corticospinal tract.
- line 255-256: In the section 3.1 you do describe CSF findings. Please clearly state here that you could not examine the CSF BEFORE functional evaluation and DTT.
Answer: It is well known that there are many blasts in the blood at the time of leukemia diagnosis. To minimize the risk of CNS contamination due to traumatic tapping, we performed CSF examination when no blast was observed in peripheral blood.
I appreciate your comment and added the sentence following your comment.
Third, we could not examine the CSF to confirm the presence of leukemic blasts to minimize the risk of CNS contamination due to traumatic tapping.

Reviewer 2 Report
the work, albeit with a limited number of case studies, puts the accept on a routine problem in clinical practice. The conclusions are interesting but need more numbers. In my opinion, the supplementary table should be added in the body of the article.
Author Response
the work, albeit with a limited number of case studies, puts the accept on a routine problem in clinical practice. The conclusions are interesting but need more numbers. In my opinion, the supplementary table should be added in the body of the article.
Answer: Thank you very much for your review.
I added the supplementary files in the main text.
